# Selection and Development of Nontoxic Nonproteolytic *Clostridium botulinum* Surrogate Strains for Food Challenge Testing

**DOI:** 10.3390/foods11111577

**Published:** 2022-05-27

**Authors:** Marijke Poortmans, Kristof Vanoirbeek, Martin B. Dorner, Chris W. Michiels

**Affiliations:** 1Department of Microbial and Molecular Systems, KU Leuven, 3000 Leuven, Belgium; marijke.poortmans@kuleuven.be (M.P.); kristof.vanoirbeel@kuleuven.be (K.V.); 2Robert Koch Institute, ZBS3-Biological Toxins, Seestr. 10, 13353 Berlin, Germany; dornerm@rki.de

**Keywords:** nonproteolytic *Clostridium botulinum*, food challenge studies, surrogate strains, nontoxic, selective medium

## Abstract

*Clostridium botulinum* causes severe foodborne intoxications by producing a potent neurotoxin. Challenge studies with this pathogen are an important tool to ensure the safety of new processing techniques and newly designed or modified foods, but they are hazardous and complicated by the lack of an effective selective counting medium. Therefore, this study aimed to develop selectable nontoxic surrogate strains for group II, or nonproteolytic, *C. botulinum*, which are psychotropic and hence of particular concern in mildly treated, refrigerated foods. Thirty-one natural nontoxic nonproteolytic strains, 16 of which were isolated in this work, were characterized in detail, revealing that 28 strains were genomically and phenotypically indistinguishable from toxic strains. Five strains, representing the genomic and phenotypic diversity of group II *C. botulinum*, were selected and successfully equipped with an erythromycin (Em) resistance marker in a defective structural phage gene without altering phenotypic features. Finally, a selective medium containing Em, cycloserine (Cs), gentamicin (Gm), and lysozyme (Ly) was developed, which inhibited the background microbiota of commercial cooked ham, chicken filet, and salami, but supported spore germination and growth of the Em-resistant surrogate strains. The surrogates developed in this work are expected to facilitate food challenge studies with nonproteolytic *C. botulinum* for the food industry and can also provide a safe alternative for basic *C. botulinum* research.

## 1. Introduction

Refrigerated processed foods with extended durability (REPFEDs) are an important and rapidly growing segment in the commercial food market. Many innovations have also been adopted in this segment, such as the use of novel protein sources, salt and sugar reduction, mild and nonthermal processing, and omission or replacement of chemical preservatives by natural alternatives [1,2]. REPFEDs are commonly processed by mild heat treatment in a temperature range of 70–95 °C, eliminating vegetative bacterial cells but not all spores. Moreover, they are often packaged under vacuum or modified atmosphere, which raises concern about the growth of anaerobic pathogens such as the sporeformer *Clostridium botulinum* [1,3]. This pathogen is notorious for producing botulinum neurotoxins (BoNTs), which, with an oral lethal dose as low as 1 ng/kg for humans, are the most potent natural toxin known [4]. BoNTs causes flaccid paralysis by blocking the release of the neurotransmitter acetylcholine at the peripheral cholinergic nerve terminals [5].

The BoNT producing *Clostridia* are divided into seven genomically and physiologically distinct species (*C. botulinum* groups I–IV, *C. sporogenes*, *C. baratii*, and *C. butyricum*), and eight major BoNT serotypes (A–G, X) have been reported [6,7,8,9]. *C. botulinum* groups I and II are primarily associated with human foodborne botulism, an intoxication caused by the consumption of food containing preformed BoNT [10,11]. Group II comprises nonproteolytic, saccharolytic strains that can produce toxin serotypes B, E, or F, and whose growth is usually prevented by a_w_ < 0.97 (corresponding to 5% NaCl), pH < 5.0, T < 3 °C [11,12]. The heat resistance of group II spores is moderate and strongly influenced by the presence of lysozyme. For example, one study reported D-values at 82.2 °C of 2.4 or 231 min in the absence or in the presence of lysozyme in the recovery medium, respectively [11]. This effect is explained by the fact that exogenous lysozyme can replace the endogenous cortex hydrolases, which are crucial for spore germination but are quite heat-sensitive [13]. Spore germination and BoNT production of group II *C. botulinum* in refrigerated foods can be controlled either by limiting the product shelf-life to <10 days [3,14], by imposing a 6-log spore reduction by heating for 10 min at 90 °C (or an equivalent time-temperature combination), or by preventing growth and toxin formation by adding preservatives, acidifying the product to pH ≤ 5, reducing the a_w_ to < 0.97, or by a combination of heat and preservative factors [15,16].

Predictive microbiology models are useful tools to assist in the design of safe food products and in the documentation of the safety of existing foods, and several models predicting the growth and toxin production of group II *C. botulinum* are currently available [17,18,19]. However, these models usually do not account for the—sometimes unexpected and pronounced—effects of the food matrix, and may thus over- or underestimate the safety risk. Therefore, unless food safety can be guaranteed by design, microbiological challenge tests should be performed to validate the safety of refrigerated foods with regard to group II *C. botulinum*. Those tests serve to assess whether new food formulations can control growth of pathogenic bacteria throughout the product shelf life, or to validate production processes intended to kill the pathogens [20]. Briefly, products are inoculated with the pathogen, incubated, and sampled over time to monitor its growth and/or inactivation. Detailed protocols for conducting challenge testing in a reproducible way have been described, also for *C. botulinum* [21]. A particular recommendation is the use of a cocktail of three to five representative strains to account for the genetic and physiological strain variability existing in the species [21].

Given that large-scale challenge testing with a highly toxic pathogen like *C. botulinum* represents a significant safety risk, the use of so-called surrogate organisms can provide an alternative [16]. The properties of *C. botulinum* surrogates should ideally be identical to those of the pathogen except for toxin production [22]. The use of such surrogates allows to identify conditions that prevent outgrowth of the pathogen and will therefore strongly reduce or eliminate the need for challenge testing with toxic strains. One possible source of nontoxic surrogate strains are natural strains of *C. botulinum* or a closely related species lacking the BoNT genes. For group I *C. botulinum*, *C. sporogenes* PA3679 has been widely used for several decades as a surrogate for the validation of thermal processes, although recent data indicate that it has a higher spore heat resistance than the (most) toxic group I *C. botulinum* strains, which may lead to overprocessing [23,24]. Additionally, recent evidence indicates that *C. sporogenes* is genetically distinct from group I *C. botulinum* based on average nucleotide identity (ANI) values, multilocus sequence typing (MLST) and core genome reconstruction [7,25,26]. For group II *C. botulinum*, in contrast, a widely used and validated nontoxic surrogate is not available [22,23,27]. One study has evaluated the potential of three naturally nontoxic isolates to serve as group II surrogates [27]. Compared to their toxic counterparts, the strains showed similar or slightly faster growth at 10 and 7 °C and over a range of different a_w_ and pH values and had similar or somewhat higher spore heat resistance. From these results, the authors concluded that these strains could serve as nontoxic surrogates for group II *C. botulinum*. However, they were poorly characterized, lacking an identification to the species level and thus having an unclear relatedness to group II *C. botulinum.* Consequently, even with the properties of the strains investigated in the paper being similar to those of toxic group II strains, it cannot be excluded that they are genetically unrelated and may show different growth, survival, or inactivation behavior in other conditions and in real foods. The study also did not investigate mutual antagonism between the strains—an undesired property for strains to be used in a cocktail. Finally, the strains had been identified as nontoxic based on the absence of detectable toxin in culture supernatant, but this does not formally exclude the possible presence of BoNT genes and the potential to produce toxin under other conditions. Besides natural nontoxic strains, an alternative source of surrogate strains are toxic strains that have been disarmed by inactivation or deletion of their toxin genes. Bradshaw et al. (2010) insertionally inactivated the BoNT/A gene in group I *C. botulinum* strain 62A with the ClosTron technology and confirmed that no phenotypic features were altered apart from toxin production [28]. Likewise, we inactivated the BoNT/E gene of the group II strain NCTC11219 with the ClosTron system and also successfully deleted the gene by allelic exchange, and the resulting nontoxic mutants had unaltered spore heat resistance and growth in unstressed and stressed conditions (NaCl, acid, and low-temperature stress) compared to the toxic parental strain [29]. Moreover, CRISPR/Cas9 technology was recently used to introduce inactivating point mutations in the *bontE* gene of group II strain Beluga, and the researchers were able to confirm the unchanged phenotype except for the production of biologically inactive BoNT/E [30].

Another hurdle that complicates challenge testing with *C. botulinum* is the lack of an effective selective medium. Botulinum selective medium (BSM), a medium containing egg yolk, cycloserine, sulfamethoxazole, and trimethoprim, was developed decades ago but does not suppress or allow to distinguish several other clostridial species and, importantly, inhibits some nonproteolytic *C. botulinum* strains [31,32]. The inclusion of antitoxin antibodies in the growth medium was shown to improve the detection of BoNT-producing strains by the formation of a precipitation zone around the colonies, but no data is available on the performance of these plates with complex samples containing background microbiota [33,34]. Finally, fluorescent antibodies against vegetative cell walls of *C. botulinum* have been used in culture, but this technique has never been adopted for use in solid media to determine *C. botulinum* cell counts [35].

The objective of this study was to overcome aforementioned problems by developing surrogate group II *C. botulinum* strains that (i) are nontoxic, (ii) have properties mimicking toxic strains, and (iii) can be easily counted on a selective plating medium. To this end, we isolated natural nontoxic group II strains, phenotypically and genotypically characterized them to select strains resembling toxic strains, inserted an erythromycin (Em) resistance gene in a defective structural prophage gene of five selected strains, and optimized a suitable selective medium for counting the surrogate strains in challenge studies. The availability of these strains and an accompanying selective medium will greatly facilitate challenge testing with group II *C. botulinum* in foods and thus stimulate innovation in the food industry.

## 2. Materials and Methods

### 2.1. Bacterial Strains, Media and Growth Conditions

Naturally nontoxic *C. botulinum* strains studied in this work include 16 own isolates (for isolation method, see Section 2.2), 12 isolates from the Robert Koch Institute (Berlin, Germany), three isolates previously described by Parker et al. (2015), and one isolate obtained from the German Collection of Microorganisms and Cell Cultures (DSM, Brunswick, Germany). Further, nontoxic *bontE* deletion mutants of *C. botulinum* NCTC8266 and NCTC11219 that our research group constructed previously were included for comparison in the physiological characterization of the naturally nontoxic isolates [29]. All these strains are listed in Appendix A, together with the *E. coli* strains used for plasmid maintenance and as conjugation donor. Media and media supplements used for growing *C. botulinum* and *E. coli* are listed in Appendix A.

All experiments described in this study were performed in a Biosafety level (BSL) 2 environment, but using BSL 3 practices when the possible presence of toxic *C. botulinum* strains (for environmental samples and enrichment cultures) or of botulinum toxin genes (for pure cultures) had not been excluded. Once the absence of the toxin genes had been demonstrated by whole genome sequencing, strains were handled using BSL2 procedures. This approach was formally approved by the Service for Biosafety and Biotechnology of the Scientific Institute for Public Health (Sciensano, document number SBB 219 2014/0018).

### 2.2. Isolation, Identification, and Whole Genome Sequencing of Nontoxic Nonproteolytic C. botulinum

Eighty-three environmental and food samples were subjected to an in-house optimized isolation protocol for group II *C. botulinum* (Figure 1). Briefly, samples were suspended in reinforced clostridial medium (RCM), subjected to a heat treatment (1 h at 67 °C) to select for sporeformers, and enriched for three days at 30 °C in an anaerobic workstation (Don Whitley A35, Don Whitley Scientific, Bingley, UK), using an atmosphere of 80% N_2_, 10% CO_2_, and 10% H_2_. Subsequently, samples were diluted and plated on RCM agar and incubated anaerobically at 30 °C for up to three days, after which morphologically different colonies were analyzed under the microscope to identify rod-shaped sporeformers. The 16S rRNA gene from selected colonies was amplified by PCR using universal primers (B27F, AGAGTTTGATCMTGGCTCAG; U1492R, GGTTACCTTGTTACGACTT) [36], the amplicons were sequenced (Macrogen Europe, Amsterdam, The Netherlands), and the results were analyzed with BLAST (https://blast.ncbi.nlm.nih.gov/Blast.cgi, accessed on 1 September 2018) and EzBiocloud (https://www.ezbiocloud.net/, accessed on 1 September 2018 [37]).

Next, genomic DNA (gDNA) was isolated from all the isolates belonging to group II *C. botulinum* based on their 16S rRNA sequence (Appendix A). In short, the cell pellet of an overnight culture in liquid RCM was treated for 1 h with PIV buffer (10 mM Tris, 1 M NaCl, pH 7.5) containing 4% formaldehyde to inhibit extracellular DNases. Afterwards, the cells were washed twice with PIV buffer to remove formaldehyde and gDNA was then extracted using a genomic DNA extraction kit (Qiagen, Hilden, Germany) and the quantity and quality of the extracted DNA was verified spectrophotometrically (mySPEC, VWR, Haasrode, Belgium; Qubit, ThermoFisher Scientific, Waltham, MA, USA) and by agarose gel electrophoresis. The gDNA was then processed through the Illumina Flex Library Prep kit (Illumina, San Diego, CA, USA) and the Nextera DNA CD Index kit (Nextera, Juno Beach, FL, USA) to prepare sequencing libraries that were run on an Illumina MiniSeq platform (Illumina, 150 bp paired-end sequencing). The reads were checked for quality, trimmed, and assembled de novo into contigs using CLC Workbench 12.0 (Qiagen). Annotation of the whole genome sequences (WGS) was performed by Rapid Annotation using Subsystems Technology (RAST Seedviewer, https://rast.nmpdr.org/, accessed on 3 January 2019) [38].

### 2.3. Genomic Characterization and Phylogenetic Analysis of the Isolates

Isolates whose WGS did not contain (parts of) a BoNT gene cluster were compared to the well-known toxic strains Alaska E43 (NC_010723.1) and Eklund 17B (NC_018648.1) by genome size and GC content (RAST Seedviewer), and by average nucleotide identity (ANI) (EzBiocloud). WGS from nine toxic strains available in GenBank (Appendix A) were additionally included in this analysis. The phylogenetic relationship between the nontoxic and toxic strains was established by multilocus sequence analysis (MLSA) with 12 housekeeping genes (Table 1) as described previously [39]. Alleles were retrieved using a BLAST search in RAST Seedviewer with the 12 genes from strain Eklund 17B (NC_018648.1), aligned using MUSCLE (https://www.ebi.ac.uk/Tools/msa/muscle/, accessed on 1 February 2019 [40]) and concatenated per strain via FASTA alignment joiner (https://birc.au.dk/~palle/php/fabox/alignment_joiner.php, accessed on 1 February 2019). A maximum likelihood phylogenetic tree was built using MEGA7. Additionally, a phylogenetic tree was established by comparing the core genome of the isolates and a set of toxic strains via Realphy (https://realphy.unibas.ch/realphy/, accessed on 1 February 2019 [41]). Multiple sequence alignments are built from these one-to-one alignments and from these a phylogenetic tree is constructed by applying the maximum likelihood method PhyML [41]. Trees were managed using the iTol online tool (https://itol.embl.de/, accessed on 1 March 2019 [42]).

### 2.4. Phenotypic Characterization

(i) Stressed growth. Salt tolerance of the isolates was determined by inoculating 0.1% of an overnight RCM broth culture in 4 mL fresh RCM broth with varying total NaCl concentrations (2.5–4.5%) and incubating anaerobically at 30°C for up to 21 days. The optical density (OD_620 nm_) was monitored daily (Ultrospec 10, Biochrom, Cambridge, UK), and cultures were considered to show growth when OD_620_ exceeded 0.15. In a similar way, acid tolerance was investigated by examining growth (OD_620_ > 0,15) for 21 days in RCM broth acidified with 1 M HCl (pH 4.94–5.25; measured after autoclaving). Growth at low temperature was evaluated by restreaking fresh 24-h colonies on RCM agar, incubating the plates at 12 °C, 7 °C, and 4 °C, and recording the time when single colonies > 1 mm diameter first appeared.

(ii) Heat resistance of spores. Spore crops were produced in a two-phase sporulation medium as described before [43]. In short, a culture grown at 30 °C from a single colony in 1 mL RCM broth was layered onto solid sporulation medium (30 mL deionized water, 3 g cooked meat medium (Thermofisher Scientific), 0.45 g agar (Neogen) and 0.03 g glucose (Thermofisher Scientific)) in a flask, together with 4 mL deoxygenized deionized water (liquid phase). After six days of anaerobic incubation, the spores were collected and washed by four cycles of centrifugation (18,000× *g*, 4 °C, 15 min) and resuspension with saline (8.5 g/L NaCl), and stored in 1 mL of saline at 4 °C. Heat resistance was measured by placing 50 µL of the suspensions in a microcentrifuge tube in a heating block at 75 °C or 85 °C for different times (1–40 min), immediately cooling them on ice, and determining plate counts on RCM agar. For the spores heated at 85 °C, the plating medium was supplemented with lysozyme (10 µg/mL) to rescue the germination mechanism of heat-damaged spores [13]. Since the first part of the inactivation curves often deviated from log-linear kinetics, decimal reduction values (D-value) were calculated from the log-linear second part of the curves, which was manually determined and included minimally four data points.

(iii) Mutual antagonism. Pairwise antagonism between the isolates was assessed using a spot halo assay. Four hundred µL of an overnight culture of every strain in RCM broth was spread on RCM agar and 5 µL of every other strain was spotted on the indicator lawn. After incubation for 24 h at 30 °C, plates were checked for inhibition halos, which reflect the production of an antimicrobial compound by the spotted strain against the indicator strain.

### 2.5. Construction of Erythromycin-Resistant Strains

The ClosTron plasmid constructs -pMTL007C-E2:XkdK-471s, pMTL007C-E2:SPP1-105s and pMTL007C-E2:TTMP523s-, harboring an intron targeted to specific genes of defective prophages (X*kdK* and *SPP1* of phage phiCT453A, and *TTMP* of phage phiCT19406A) were designed via the ClosTron design tool (http://clostron.com/clostron2.php, accessed on 1 September 2019 [44]), transformed via electroporation into *E. coli* DH5α for maintenance, and into *E. coli* S17 λpir for conjugation to *C. botulinum*. The sequence of the target genes and of the retargeted intron are provided in Appendix A. *E. coli* S17 λpir carrying a ClosTron plasmid construct (pMTL007C-E2:XkdK-471s, pMTL007C-E2:SPP1-105s or pMTL007C-E2:TTMP-523s) was grown overnight in lysogeny broth (LB) with chloramphenicol (Cm), washed twice by centrifugation (6700× *g*, 5 min, 4 °C). Subsequently, the pellet was brought into the anaerobic cabinet and resuspended in 200 µL of an overnight culture of the recipient nontoxic *C. botulinum* isolate in RCM broth. This mixture was spread onto a 0.45 µm filter (Sigma Aldrich, Saint-Louis, MO, USA) on RCM agar. After 24 h of anaerobic incubation at 30 °C, the cells were recovered in 1 mL saline, and the mixture was plated on RCM agar with cycloserine (Cs) and thiamphenicol (Tm) to counterselect the *E. coli* donor and to select for conjugants bearing the ClosTron plasmid, respectively. Colonies were purified and the presence of the plasmid was confirmed by PCR with primers for the *catP* gene (Table 2). Purified conjugants were subsequently streaked onto RCM agar with erythromycin (Em, 5 µg/mL) to select for colonies in which the ClosTron intron and the embedded retrotransposition-activated marker (RAM) based on the ErmB gene, had been inserted into the genome. After purification of the resulting colonies, correct insertion of the intron was confirmed by colony PCR and sequencing of the *Xkdk, SPP1 or TTMP* target genes (Table 2). Finally, selected colonies were verified to have lost the ClosTron plasmid by loss of Tm resistance and by PCR targeting the *catP* gene. WGS analysis of the mutants was conducted to confirm the absence of adventitious mutations.

To study the stability of the introduced Em marker, overnight cultures of the Em-resistant mutants, in triplicate, were re-inoculated (1:1000) into fresh RCM broth every 24 h for 7 days, after which a plate count on RCM agar with and without Em (5 µg/mL) was conducted.

### 2.6. Selective Medium for C. botulinum Surrogate Strains

A selective medium for food challenge studies with the constructed Em-resistant *C. botulinum* surrogates was composed by supplementing RCM with different combinations of antibiotics, and with lysozyme to stimulate spore germination. The effectiveness of the media to suppress background microbiota was tested with three commercial meat products (Table 3).

## 3. Results and Discussion

### 3.1. Isolation and Genomic and Phylogenetic Analysis of Nontoxic Nonproteolytic C. botulinum Strains

From the 83 collected food and environmental samples, 16 strictly anaerobic spore-forming isolates clustering with toxic group II *C. botulinum* strains based on 16S rRNA sequence similarity were isolated. The whole genome sequences (WGS) of these isolates and of 15 isolates from other sources (Appendix A) were determined and annotated, and all were found to lack a BoNT toxin gene cluster. The WGS were then compared to 11 publicly available WGS from toxic group II strains (Table 4).

ANI analysis shows that 26 nontoxic strains share more similarity (>97%) with toxin type B strain Eklund 17B than with toxin type E strain Alaska E43 (<94%), while the opposite is true for two strains (ZBS3 and DSM1985). Eklund 17B and Alaska E43 were chosen because they are representative for the previously identified type BEF and type E clusters of group II *C. botulinum*, respectively [7,39,45]. Furthermore, three strains (ZBS2, ZBS15, and CMCC3677) show low similarity (<93%) with either of the toxic strains, but also with other *C. botulinum* groups or closely related species such as *C. perfringens, C. butyricum,* and *C. baratii* (data not shown). The genome size and GC content of all 31 nontoxic strains are in the same range.

Phylogenetic analysis of the 31 nontoxic strains and 11 toxic strains (WGS from NCBI), based on 12 household genes (Figure 2A) and on the core genomes (Figure 2B), confirmed the existence of the previously identified type E and type BEF cluster. The BEF cluster itself consists of two subclusters, one containing toxin type B4 and F6 strains as well as nontoxic strains, and the other lineage containing toxin type B4 and E and nontoxic strains [45]. The presence of nontoxic and toxic strains in all these clusters and subclusters indicates that nontoxic and toxic group II *C. botulinum* are phylogenetically indistinguishable (Figure 2A,B). The phylograms based on housekeeping gene analysis and core genome analysis also both show two BEF subclusters, similar to the findings of Brunt et al. (2020b), but the strains are not always consistently mapped to these subclusters. For example, strain ZS1 is in subcluster one based on the housekeeping gene phylogram, and in subcluster two based on the core genome phylogram. The phylograms also confirm the finding from ANI analysis that strains ZBS2, ZBS15, and CMCC3677 constitute a separate lineage that is unrelated to group II *C. botulinum.* CMCC3677, together with CMCC3676 and CMCC3678, was previously proposed to be a suitable nontoxic surrogate strain for group II *C. botulinum* based on phenotypic analysis, but our genomic analysis clearly isolates it from the two other strains. This makes it conceivable that this strain may possess phenotypic traits that have not been revealed but may render it unsuitable as a surrogate for food challenge studies, thus underscoring the importance of genomic analysis for identifying nontoxic surrogate strains for food challenge studies. 

### 3.2. Growth in High Salt, at Low pH and at Low Temperature

Growth of the nontoxic group II *C. botulinum* strains under salt, acid, and cold stress was determined in comparison to two toxic strains in which the toxin gene had been deleted and replaced by an Em resistance marker (NCTC8266Δbont::ermB and NCTC11219Δbont::ermB).

The maximum NaCl concentration allowing growth under the applied test conditions ranged from 2.5 to 4.0%, depending on the strain, but also with large differences in the time to growth (OD_620_ > 0.15) (Table 5). The most salt-tolerant strain (ZBS3) was able to grow at 4% NaCl in 2 days but was nevertheless not able to grow at 4.5% NaCl. In acidified RCM broth, growth of all the strains except NCTC11219Δbont::ermB was inhibited at pH 4.94, and the minimum pH supporting growth was 5.05 or 5.12 depending on the strain (Table 6). Finally, the temperature limiting growth was 4 °C or 7 °C depending on the strain (Table 7). Growth at 4 °C was always slow, with 20–21 days required to observe 1 mm colonies. The observed pH and temperature limits of the nontoxic strains generally correspond to the range described in the literature for toxic group II strains. On the other hand, the maximum salt tolerance found in our work appears to be lower than the 5% maximum found in some reports [1,11]. However, some studies also report lower experimental maximum values, and these differences may relate to the different incubation times used [16,46]. Possibly, extending the incubation time could result in higher salt tolerance values for our strains as well.

Strains from the two group II genomic subclusters showed no notable differences in acid, salt and cold tolerance. In contrast, strains from the non-group II cluster—ZBS2, ZBS15, and CMCC3677—showed higher salt and lower cold tolerance than most genuine group II strains, supporting the hypothesis that this group is not only genetically but also phenotypically distinct from group II *C. botulinum*. In a previous genomic and physiological study of nonproteolytic strains, the fermentation of different carbohydrates was found to be cluster-specific. Strains from the BEF cluster produced acid from amylopectin, amylose, and glycogen, but not from melezitose, while cluster E strains only fermented the latter [12]. Therefore, it is recommended to select strains from both the type E cluster and the type BEF cluster for challenge studies, since different characteristics may result in different growth patterns in food products [10].

Besides strain-to-strain variation, considerable variation between replicate cultures of the same strain was sometimes observed in these tests (Appendix A). This variation may be due to the large variation in lag phase between individual cells that exists particularly in stressed growth conditions, and that is believed to be advantageous to ensure population survival [47].

### 3.3. Spore Heat Resistance

It has been well documented that the apparent heat resistance of group II *C. botulinum* spores is strongly increased when lysozyme is present in the recovery medium [13,43]. This is because the endogenous spore cortex hydrolases, which are essential for spore germination, are very heat-sensitive, but can be replaced by exogenous hydrolases. Therefore, spore heat resistance was analyzed at 75 °C in the absence, and at 85 °C in the presence of lysozyme.

At 75 °C, many inactivation curves showed a shoulder or even an activation phase that suggests the presence of a superdormant spore fraction, as has been previously observed for spores of several bacilli and clostridia (Appendix A) [48]. Superdormant fractions comprised up to 99% of the spore population in some strains (KI2, ZBS17, RO132). The D_75_ values, calculated from the log-linear part of the inactivation curves, showed a broad variation, ranging from 1.8 to 21.4 min (Figure 3), but the average D_75_ (7.4 min) corresponded well to the value reported in a meta-analysis of literature data for toxic group II strains (5.1 min) [49]. Furthermore, the four strains of the type E cluster included in our experiment (ZBS3, DSM1985, NCTC8266 Δbont::ermB and NCTC11219 Δbont::ermB) had the lowest spore heat resistance of all the strains in the absence of lysozyme (D_75_ < 3 min), confirming a similar finding from the same meta-analysis.

At 85 °C, the addition of lysozyme resulted in biphasic reduction curves with a rapid inactivation phase followed by a slower phase, probably because the spore coat in a fraction of the spore population is impermeable to lysozyme and this fraction can therefore not be rescued [50]. D-values at 85 °C were therefore determined from the heat-resistant (lysozyme-rescued) spore fraction only. The average D-value of the nontoxic isolates at 85 °C in the presence of lysozyme was 19.8 min (range from 9.1 to 40.7 min), which is somewhat lower than the corresponding value obtained for toxic strains under similar conditions (33.1 min) obtained in the meta-analysis of Wachnicka et al. (2016). The reason for this difference is not clear but may be related to the wide variation in experimental conditions underlying the data in the meta-analysis, which included studies conducted in food matrices and using different concentrations of lysozyme in the recovery medium. Interestingly, while the spores of the type E cluster strains were the most heat sensitive in the absence of lysozyme, their average heat resistance in the presence of lysozyme (average D_85_ = 27.8) was higher than that of the type BEF cluster strains (average D_85_ = 18.7). This may indicate that type E cluster strains have more heat-sensitive cortex hydrolases.

Finally, two of the three strains from the separate non-group II phylogenetic cluster (ZBS15, CMCC3677, but not ZBS2) showed high spore heat resistance both at 75 °C and with lysozyme at 85 °C.

### 3.4. Mutual Antagonism between Nontoxic Isolates

Since it is the intention to combine selected nontoxic *C. botulinum* strains into multi-strain cocktails for conducting food challenge tests, it is important that they do not inhibit each other by the production of antimicrobial compounds. Therefore, the pairwise mutual inhibition of all the strains was systematically analyzed in a plate halo assay (Table 8). The majority of strains (23) produced no or only very small halos against all the other strains. Very small halos were neglected because they were considered more likely to be the result of nutrient competition than from the production of specific antimicrobial substances in this assay. Eight strains produced a halo of moderate size against at least one other strain. Here, production of an antimicrobial compound cannot be excluded but should be further confirmed. Finally, two strains produced a large halo against at least one other strain. ZBS20 produced a large halo against one and a moderate halo against seven other strains, and DSM1985 produced a large halo against 24 and a moderate halo against eight strains. The strong antagonism of DSM1985 (also designated as strain S5 in other studies) against other nonproteolytic *C. botulinum* strains has been reported before, and production of inhibitory substances has been documented also in other strains of both nonproteolytic and proteolytic *C. botulinum* [51,52,53]. It should be noted that the absence of detectable antimicrobial activity in the halo assay does not exclude the possibility that a strain may still produce such activity in other conditions, but this was not further investigated in our work.

### 3.5. Selection of Strains for Challenge Testing and Introduction of an Erythromycin Resistance Marker

The data obtained from the genomic and phenotypical characterization were used to select a set of five strains for inclusion in a cocktail deemed to be suitable for food challenge studies. Since challenge studies are aimed at identifying potential microbiological risks, it is important that the selection includes at least one strain that has a high tolerance to growth inhibition by low pH, low temperature, and high salt concentration, respectively, and one strain with highly heat-resistant spores. Furthermore, both phylogenetic lineages should be represented because these may differ in some growth or survival-related properties that were not tested in this work but that may be relevant in specific foods. Finally, the selected strains should not show cross-inhibition. Based on these considerations, five strains were selected (Table 9). ZBS3 was selected to represent the type E cluster, since the only other nontoxic strain from this cluster (DSM1985) was antagonistic to most other strains. ME22 and VAP51 were included for their good growth at low temperature, ZBS4 for its salt tolerance, and CH2 for its tolerance to low pH and the heat resistance of its spores.

To facilitate selective counting of the selected strains in food challenge tests, an Em resistance gene was inserted in their chromosome using the ClosTron system [44]. Insertions were targeted to structural genes of defective prophages, since it was anticipated that this would be unlikely to modify any relevant phenotype. Since the five strains did not share a common defective prophage structural gene, three different targets were used: gene *XkdK* from the non-intact phage phiCT453A for strains ME22, VAP51 and ZBS4, gene *SPP1* from the same phage for strain ZBS3, and gene *TTMP* from prophage phiCT19406A for strain CH2. After insertion of the ClosTron intron in these phage genes, Em-resistant mutants that had lost the Clostron delivery plasmid were identified by PCR. Loss of this plasmid is important since this precludes the LtrA-mediated splicing of the intron from the target sequence and is thus expected to enhance the stability of Em resistance. In addition, WGS analysis confirmed that the Em-resistant mutants had not picked up any adventitious mutations. Finally, the growth under salt, acid, and cold stress and the spore heat resistance of the Em-resistant mutants was re-evaluated and found to be indistinguishable from their respective parental strains (data not shown). Based on these data, it was concluded that well-targeted ClosTron insertion can be used for generating antibiotic-resistant *C. botulinum* strains without altering important phenotypical features.

### 3.6. Stability of Erythromycin Resistance

Intron integrants such as ClosTron mutants are intrinsically stable [44]. However, the prophage regions into which the introns were targeted in this work, despite being predicted to be defective, could potentially be unstable. The stability of the introduced Em resistance marker was therefore determined by growing three independent cultures of the five constructed ClosTron mutants for approximately 70 generations without Em, and subsequent plate counting on medium with and without Em. No significant differences in the plate counts on the two media were found for any of the cultures, indicating that the introns carrying the Em resistance marker are stable in all the mutants (data not shown).

### 3.7. Selective Medium for Challenge Studies

The insertion of an Em resistance marker is expected to facilitate the counting of the modified nontoxic *C. botulinum* strains in food challenge experiments because it allows to include Em in the plating medium to increase its selectiveness, in particular against other clostridial species which are naturally susceptible to Em [54,55,56]. Here, a selective medium for use with the Em-resistant challenge strains is composed and its effectiveness is tested.

In a first experiment, different selective media containing Em were evaluated to compare their effectiveness to suppress the background microbiota of cooked ham, chicken filet, and salami, which amounted to 6.5 ± 0.1 log cfu/g, 3.3 ± 0.9 log cfu/g and 7.0 ± 0.04 log cfu/g, respectively, on RCM agar upon anaerobic incubation. Em was added at 3 µg/mL, a concentration at which the Em-resistant ClosTron mutants still showed 100% plating efficiency. Since this concentration of Em in the RCM plating medium only partially inhibited the endogenous microbiota (Figure 4), Cs and Gm were used as additional antibiotics to identify a combination that was more effective against the background (Figure 4). Use of these antibiotics to select and isolate *C. botulinum* has been previously proposed, based on the natural levels of resistance of this organism against Cs (> 256 µg/mL) and Gm (87% of the tested strains was resistant to 16 µg/mL, 95% to 8 µg/mL) [55,56,57]. Suitable concentrations of Cs and Gm that did not reduce the plating efficiency of the five ClosTron mutants were determined to be 100 µg/mL and 15 µg/mL, respectively. When used alone, Cs and Gm were generally less effective than Em to suppress the background microbiota of the three tested meat products. Pairwise combinations of antibiotics were more suppressive, but none of the combinations was sufficiently effective on all food products, and particularly the background of the cooked ham was insufficiently inhibited. However, the combination of all three antibiotics completely eliminated the background of all three products, except for one colony in one of the three replicates of cooked ham. In conclusion, RCM + Em + Cs + Gm is very effective to suppress the endogenous microbiota of these cooked meat products, even at high levels (Figure 4).

In a second experiment, the plating efficiency of the five Clostron mutant strains on RCM + Em + Cs + Gm compared to RCM without antibiotics was evaluated. This was done for each strain separately and with the five-strain cocktail, and both with spore suspensions and suspensions of vegetative cells (Figure 5). The plating efficiency of the vegetative cells on the non-selective and selective medium was indistinguishable for each of the strains and the cocktail. In contrast, counts of the spore suspensions were 0.5–1.7 log lower on the selective medium, depending on the strain, although the differences were not always significant at the 5% level. Prolonged incubation did not increase the counts on the selective medium. The lower plating efficiency specifically of the spores suggests that the antibiotic cocktail somehow interferes with spore germination, and therefore the selective medium was additionally supplemented with lysozyme in an attempt to stimulate spore germination. The results in Figure 5 show that lysozyme indeed increased the plating efficiency of the spore suspensions for all the strains. For three strains, the plating efficiency was restored to the level of the non-selective medium, for one strain it was still slightly lower (0.4 log), and for one strain it was even higher, which might be due to a superdormant fraction that does not germinate even on RCM without the help of exogenous lysozyme

## 4. Conclusions

This work reports a detailed phenotypic and genomic analysis of 31 nontoxic nonproteolytic *C. botulinum* strains with the aim of developing a set of surrogate strains for challenge testing in foods. The genomic analysis based on the strains’ WGS revealed three lineages; the previously described type E and type BEF clusters of toxic nonproteolytic strains, and a novel, quite distinct and yet uncharacterized cluster. The strains’ capacity to grow under salt, acid, and low temperature stress, and their spore heat resistance were determined and shown to fall in the same range as described for toxic nonproteolytic strains. Together, these results show that, apart from the presence of a BoNT toxin gene cluster, nontoxic, and toxic nonproteolytic Group II *C. botulinum* strains are genetically and phenotypically indistinguishable. Five nontoxic strains, encompassing the natural genomic and phenotypic diversity present in group II *C. botulinum*, were equipped with a genomic Em resistance marker, and a selective medium that suppresses the background microbiota of three commercial meat products and allows quantitative recovery of the Em-resistant surrogate strains, was developed. Although they still need to be validated in full-scale challenge studies, the surrogate strains and the accompanying selective medium developed in this work can be anticipated to make food challenge testing with nonproteolytic *C. botulinum* safer, easier, and more reliable, and thus more accessible to the food industry. It is important to note that toxic strains may still need to be used, because guidelines in some countries require neurotoxin-based challenge testing for *C. botulinum*. However, even in such cases, prior testing with the surrogates will allow to reduce the design and cost of challenge testing with toxic strains. Finally, the availability of a collection of nontoxic strains that is otherwise indistinguishable from toxic strains also paves the way for use of these strains as a safe alternative in basic research on nonproteolytic *C. botulinum*, and this will hopefully promote a better understanding of the genetics, physiology, and ecology of these interesting bacteria.

## Figures and Tables

**Figure 1 foods-11-01577-f001:**
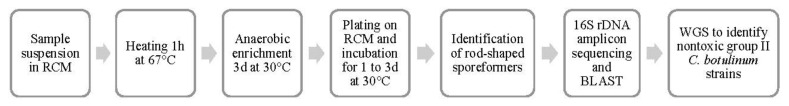
Overview of the isolation procedure for nontoxic nonproteolytic *C. botulinum*.

**Figure 2 foods-11-01577-f002:**
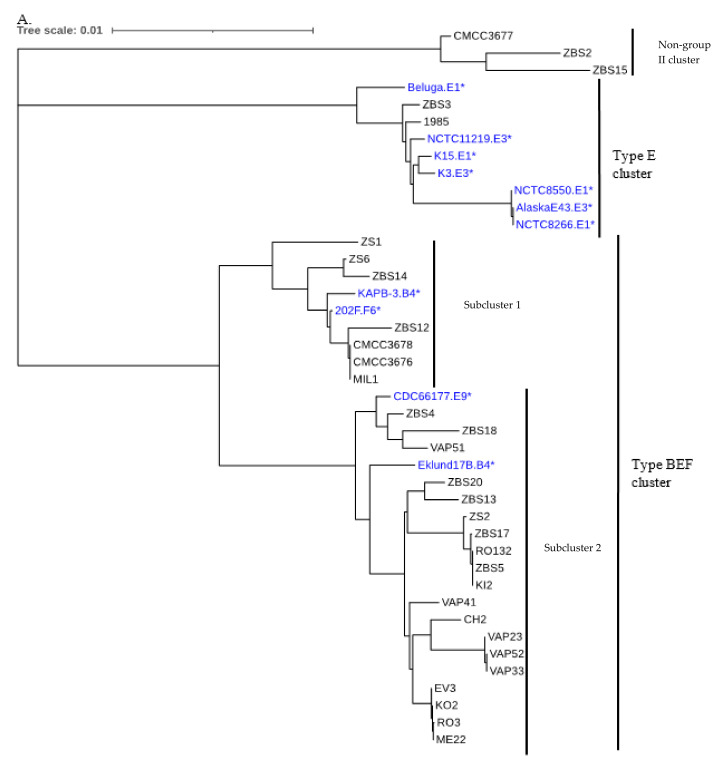
Phylogenetic trees of group II and non-group II nontoxic and toxic strains: (**A**) Analysis based on 12 housekeeping genes. (**B**) Analysis based on core genomes. Black, WGS determined in this work; Blue, WGS from NCBI. Toxic strains: * Scale indicates the number of substitutions per nucleotide site.

**Figure 3 foods-11-01577-f003:**
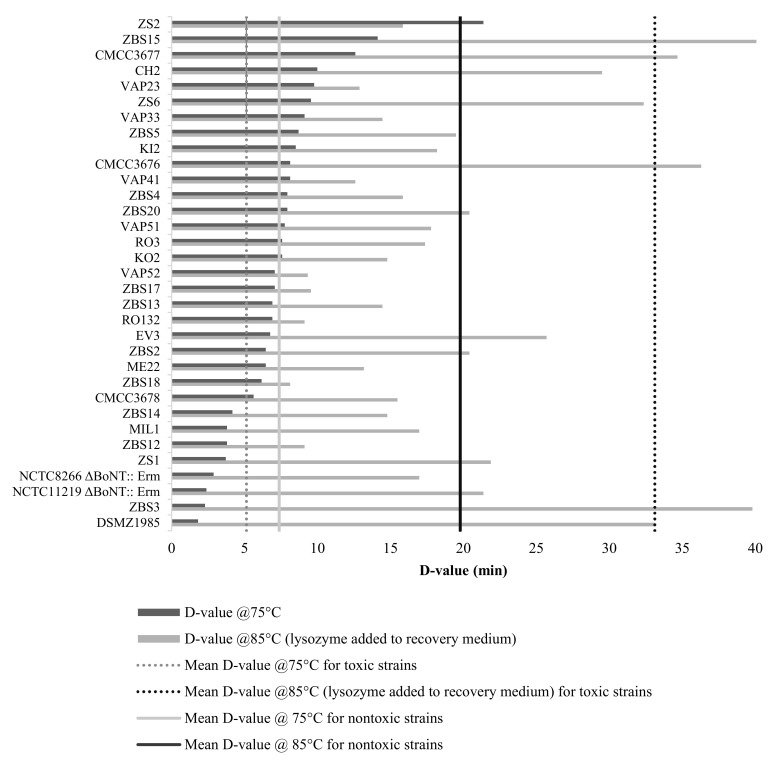
D-values at 75 °C and 85 °C for all nontoxic isolates, NCTC8266Δbont::ermB and NCTC11219Δbont::ermB. The recovery medium for spores treated at 85 °C was supplemented with 10 µg/mL lysozyme. Vertical dashed lines represent mean D-values for strains analyzed in this work at 75 °C (light grey) and at 85 °C (black). Vertical dotted lines represent mean D-values for toxic group II strains from a published meta-analysis at 75 °C (light grey) and at 85 °C with lysozyme (black). Reprinted/adapted with permission from Ref. [49]. Copyright 2016, copyright Wachnicka et al. D-values were calculated from the log-linear part of the reduction curve.

**Figure 4 foods-11-01577-f004:**
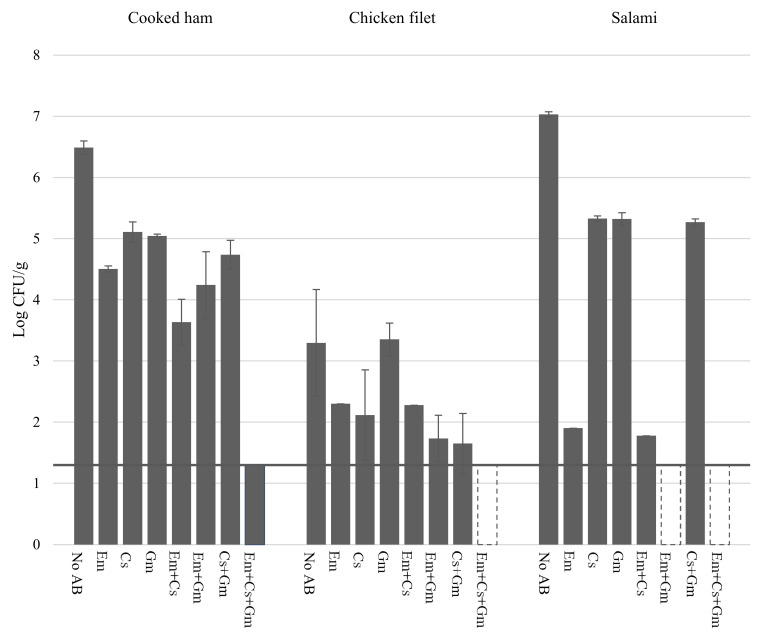
Anaerobic plate counts of background bacteria from cooked ham, chicken filet, and salami on RCM agar, and RCM agar with different combinations of the antibiotics Em (3 µg/mL), Cs (100 µg/mL) and Gm (15 µg/mL). Horizontal line indicates lower detection limit. Bars in dashed lines indicate complete absence of colonies. Error bars represent standard deviation of three independent samples from the same food packaging unit.

**Figure 5 foods-11-01577-f005:**
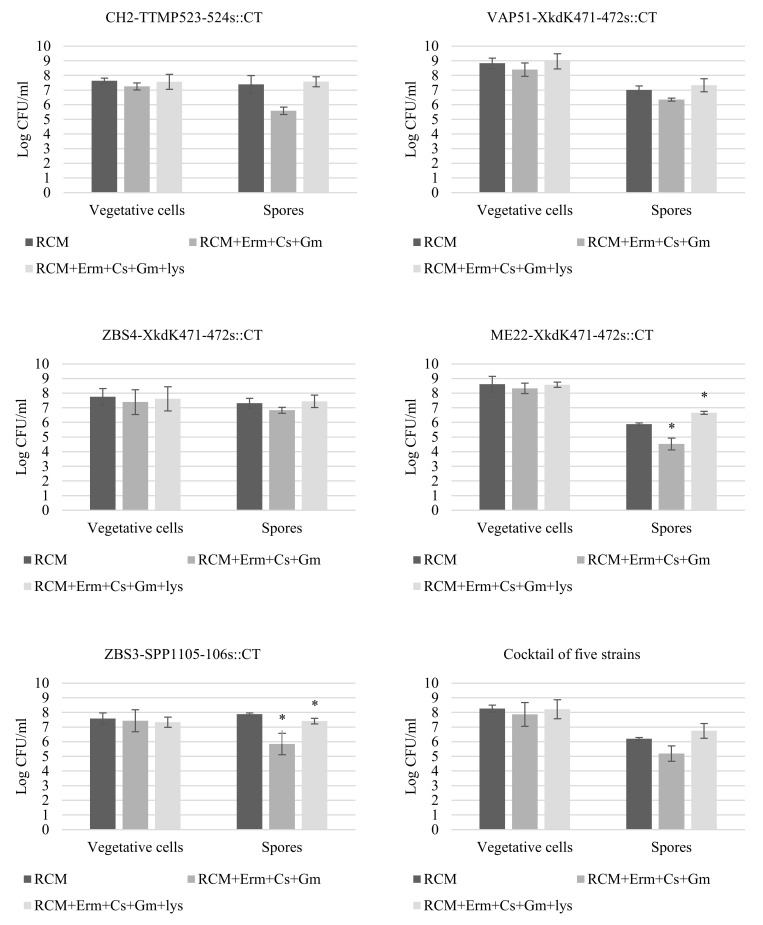
Anaerobic plate counts of suspensions of spores and vegetative cells of the five Em-resistant ClosTron mutants, separately and together in a cocktail, on RCM agar, RCM agar + Em (3 µg/mL) + Cs (100 µg/mL) + Gm (15 µg/mL), and RCM agar with the three antibiotics and lysozyme (lys, 50 µg/mL). Error bars represent standard deviations of three independent overnight cultures/spore suspensions. Significant differences (*, *p* < 0.05) between counts on the selective media and RCM agar were calculated using a one-way ANOVA and a paired *t*-test.

**Table 1 foods-11-01577-t001:** Genes used in MLSA of nonproteolytic *C. botulinum* strains. Reprinted/adapted with permission from Ref. [39]. Copyright 2015, copyright Weedmark et al.

Gene	Product
atpD	ATP synthase subunit beta
guaA	GMP synthase (glutamine-hydrolyzing)
gyrB	DNA gyrase subunit beta
ilvD	Dihidroxy-acid dehydratase
IepA	Elongation factor A
oppB	Oligopeptide transport system permease
rpoB	RNA polymerase subunit beta
trpB	Tryptophan synthase beta chain
recA	DNA recombination and repair protein
pyc	Pyruvate carboxylase
Pta	Phosphate acetyltransferase
23S	23S rRNA

**Table 2 foods-11-01577-t002:** Primers used for *CatP, XkdK, SPP1,* and *TTMP* amplification.

Primer	Sequence (5′-3′)
CatP_F	AAGGAAAGCCAAATGCTCCG
CatP_R	ACGGCAAATGTGAAATCCGTC
XkdK_F	GCTACGAAGGTGCTAGGAGA
XkdK_R	ATCCTGCTGTTAATGCCGCTA
SPP1_F	GGAGGCGGTATGTTGGGAG
SPP1_R	ACCTTTGTTGCTTGCCTCAT
TTMP_F	TATATGGCGATGGCGGGTTG
TTMP_R	TTTCCGAAAAGTGTTGCGGC

**Table 3 foods-11-01577-t003:** Characteristics of commercial meat products used for testing inhibition of background microbiota by selective medium for Em-resistant *C. botulinum* strains.

Product	‘Use by’ Date	Purchase Date
Cooked ham	29 October 2021	25 October 2021
Chicken filet	9 November 2021	25 October 2021
Salami	30 November 2021	25 October 2021

**Table 4 foods-11-01577-t004:** Genomic characterization of nontoxic and toxic nonproteolytic *C. botulinum*. ANI was calculated with toxic strains Alaska E43 and Eklund 17B, representative of the previously identified type E and type BEF clusters of group II *C. botulinum*. Toxic strains: *.

Strain	Toxin Type	Genome Size (bp) (Sum of Contigs)	GC Content (%)	ANI (Alaska E43)	ANI (Eklund 17B)
**WGS determined in this work ^a^**					
KI2	-	3,767,496	27.2	93.51%	98.81%
ME2.2	-	3,911,728	27.2	93.30%	98.76%
CH2	-	3,689,533	27.2	93.45%	98.81%
ZS1	-	3,696,074	27.2	93.61%	97.75%
ZS2	-	3,772,516	27.2	93.40%	98.79%
ZS6	-	3,629,812	27.1	93.78%	97.71%
EV3	-	3,911,500	27.2	93.58%	98.81%
KO2	-	3,912,807	27.2	93.50%	98.74%
RO3	-	3,911,988	27.2	93.38%	98.77%
RO132	-	3,767,657	27.2	93.50%	98.85%
MIL1	-	3,884,748	27.2	93.70%	97.61%
VAP23	-	3,679,375	27.1	93.70%	98.89%
VAP33	-	3,668,189	27.2	93.55%	98.89%
VAP41	-	3,727,139	27.2	93.38%	98.98%
VAP51	-	3,759,828	27.2	93.48%	98.86%
VAP52	-	3,844,223	27.1	93.54%	98.85%
ZBS2	-	3,685,079	27.2	91.12%	92.09%
ZBS3	-	3,657,686	27.1	98.86%	93.73%
ZBS4	-	3,812,273	27.2	93.70%	98.90%
ZBS5	-	3,825,512	27.2	93.47%	98.78%
ZBS12	-	3,805,576	27.2	93.51%	97.71%
ZBS13	-	3,777,964	27.1	93.34%	98.75%
ZBS14	-	3,753,809	27.2	93.61%	97.74%
ZBS15	-	3,685,017	27.3	91.33%	92.34%
ZBS17	-	3,798,380	27.3	93.48%	98.89%
ZBS18	-	3,922,262	27.1	93.56%	99.11%
ZBS20	-	3,734,895	27.3	93.40%	99.09%
DSM1985	-	3,830,944	27.6	99.18%	93.67%
CMCC3676	-	3,913,751	27.3	93.73%	97.56%
CMCC3677	-	3,647,744	27.4	91.19%	92.11%
CMCC3678	-	3,923,472	27.3	93.68%	97.59%
**WGS from NCBI**					
K3 *	E3	3,850,230	27.1	99.38%	93.56%
K15 *	E1	3,997,940	27.2	98.79%	93.56%
CDC66177 *	E9	3,852,440	27.2	93.70%	99.05%
KAPB-3 *	B4	3,871,080	27.3	93.78%	97.63%
NTCT8550*	E1	3,611,898	27.4	99.97%	93.58%
Beluga *	E1	3,863,095	27.3	97.96%	93.58%
202F *	F6	3,874,462	27.4	93.46%	97.62%
NCTC8266 *	E1	3,661,134	27.1	99.97%	93.59%
NCTC11219 *	E3	3,792,090	27.4	99.26%	93.69%
Eklund 17B *	B4	3,800,327	27.5	93.62%	100%
Alaska E43 *	E3	3,659,644	27.4	100%	93.62%

^a^ Isolates with 16S sequence identity > 98% with group II *C. botulinum* strains available in the EzBiocloud and NCBI database, and <97% with other clostridial species.

**Table 5 foods-11-01577-t005:** Growth of nontoxic isolates, NCTC8266Δbont::ermB and NCTC11219Δbont::ermB at 30 °C in RCM broth with 2.5%–4.5% NaCl. Growth was monitored for 21 days for three replicates per strain. None of the strains grew at 4.5% NaCl.

Days to Growth (OD_620_ > 0.15) at Different Salt Concentrations for at Least One Replicate									
Strain	NaCl%				Strain	NaCl%			
	2.5	3.0	3.5	4.0		2.5	3.0	3.5	4.0
ZBS3	1	1	2	2	RO132	1	1	6	
ZS6	1	1	2	6	DSM1985	2	1	6	
ZBS5	1	2	6	7	VAP23	1	2	6	
ZBS4	1	1	6	7	NCTC8266 Δbont::ermB	1	2	6	
CMCC3677	1	1	2	8	ZBS20	1	2	6	
VAP51	1	1	13	16	VAP41	1	2	7	
VAP52	1	1	2		CMCC3678	1	2	8	
ZBS15	1	2	2		ZS1	1	2	8	
ZBS14	1	2	2		ME22	1	5	13	
ZBS2	1	2	2		RO3	1			
CH2	1	1	6		KO2	1			
ZS2	1	1	6		EV3	1			
VAP33	1	1	6		ZBS13	1			
KI2	1	1	6		ZBS12	1			
ZBS17	1	1	6		MIL1	1			
NCTC11219 Δbont::ermB	1	1	6		CMCC3676	2			
ZBS18	1	1	6						

**Table 6 foods-11-01577-t006:** Growth of nontoxic isolates, NCTC8266Δbont::ermB and NCTC11219Δbont::ermB in RCM broth acidified with 1 M HCl to pH 4.94–5.25. Growth was monitored for 21 days for three replicates per strain.

Days to Growth (OD_620_ > 0.15) at Different pH Levels for at Least One Replicate									
Strain	pH				Strain	pH			
	4.94	5.05	5.12	5.25		4.94	5.05	5.12	5.25
NCTC11219 Δbont::ermB	5	4	2	1	ZS2			2	1
VAP23		5	2	1	ZBS12			2	1
CH2		5	2	1	ZS1			2	1
VAP33		5	2	1	ZBS3			2	1
EV3		5	2	1	DSM1985			2	1
KO2		5	3	1	ZBS20			2	1
RO3		6	2	1	ZBS18			2	1
RO132		7	2	1	ZBS2			2	2
ME22		7	2	1	NCTC8266 Δbont::ermB				1
ZBS4		7	2	1	ZBS14				1
CMCC3677		9	2	1	ZBS13				1
KI2		9	2	1	ZS6				1
ZBS5		12	2	1	CMCC3676				5
ZBS17		12	2	1	CMCC3678				5
VAP52		17	2	1	ZBS15				5
VAP51			2	1	MIL1				5
VAP41			2	1					

**Table 7 foods-11-01577-t007:** Growth of nontoxic isolates, NCTC8266Δbont::ermB and NCTC11219Δbont::ermB on RCM agar at 4 °C, 7 °C and 12 °C, monitored for 21 days for three replicates per strain.

Days to Colony Formation (>1 mm) at Different Temperatures for at Least One Replicate							
Strain	Temperature			Strain	Temperature		
	4 °C	7 °C	12 °C		4 °C	7 °C	12 °C
ZBS14	20	8	3	KI2		7	3
ZS6	20	9	3	VAP33		7	3
RO132	20	9	3	VAP52		7	3
ZBS3	20	10	3	CMCC3676		9	3
ZS1	20	14	3	ZBS12		9	3
VAP41	20	7	3	VAP23		9	3
EV3	20	7	7	NCTC8266 Δbont::ermB		10	2
ME22	20	7	3	NCTC11219 Δbont::ermB		10	2
VAP51	20	7	1	ZBS20		10	3
RO3	20	9	2	ZBS5		11	3
DSM1985	20	14	3	KO2		14	3
MIL1	21	9	1	CH2		14	3
ZBS18		7	1	CMCC3677		14	3
ZBS17		7	1	ZBS15		15	3
CMCC3678		7	3	ZBS4		15	3
ZBS13		7	3	ZBS2		21	3
ZS2		7	3				

**Table 8 foods-11-01577-t008:** Mutual antagonism between all nontoxic isolates, NCTC8266Δbont::ermB and NCTC11219Δbont::ermB based on plate halo assay. Halo size: +: small (<1 mm), ++: medium (1–3 mm), +++: large (>3 mm).

Producer Strain																																	
Indicator Strain	KI2	ME2.2	CH2	ZS1	ZS2	ZS6	EV3	KO2	RO3	RO132	MIL1	VAP23	VAP33	VAP41	VAP51	VAP52	ZBS2	ZBS3	ZBS4	ZBS5	ZBS12	ZBS13	ZBS14	ZBS15	ZBS17	ZBS18	ZBS20	DSM1985	CMCC3676	CMCC3677	CMCC3678	NCTC8266	NCTC11219
KI2																+											+	+++					
ME22														+								+					+	++					
CH2														+					+								+	+++					
ZS1														+					+			+	+				+	+++					
ZS2	+									+						+				+							+	+++					+
ZS6	+	+					+			+				+					+			+	+				+	++					+
EV3																											+	++					
KO2														+													+	++					
RO3																						+						+++					
RO132																											+	+++					
MIL1	+		+		+					++			+	+		+				+		+	+				+++	+++			+		
VAP23	+		+		+					+				+						+							++	+++					
VAP33	+									+				+					+				+				++	+++					
VAP41																											+	+++					
VAP51																											++	+++					
VAP52																			+			+					+	+++					
ZBS2														+					+			+	++				+	+++		+	+		
ZBS3	+	+								++				++					+			+	+				++	++			+		
ZBS4																												+++					
ZBS5						+							+	+		+											+	+++					
ZBS12	+																		+			+	+				+	+++					
ZBS13	+				+		+	+		+				+		+				+							+	+++				+	+
ZBS14	+				+					+			+	+		+			+	+							+	+++					
ZBS15																			+			+	+				+	+++					
ZBS17														+								+	+					+++				+	+
ZBS18	+									+				+					+	+							+	+++					
ZBS20																												++					
DSM1985						+		+						+					+				+				+						
CMCC3676	+		+				+			+				++		++				++	+	+					++	+++				+	+
CMCC3677	+	+	+	+		+	+	+						++					++				+				++	+++					
CMCC3678	+		+		++			+		++			++	++		++				++		+	+				++	+++				+	+
NCTC8266						+			+					+			+		+				+				+	++					
NCTC112198	+				+	+				+				+						+		+	+		+		+	++	+				

**Table 9 foods-11-01577-t009:** Summary of characteristics of nontoxic strains selected for inclusion in a challenge testing strain cocktail. D_85_ was determined with lysozyme in the plating medium.

Strain	Phylogenetic Cluster	Max NaCl (%) (Days to Growth)	Min pH (Days to Growth)	Min T (°C) (Days to Growth)	D_75_-Value (min)	D_85_-Value (min)
CH2	BEF	3.5 (6)	5.05 (5)	7 (14)	10	29.5
VAP51	BEF	4 (16)	5.12 (2)	4 (20)	7.8	17.8
ZBS4	BEF	4 (7)	5.05 (7)	7 (15)	7.9	15.9
ME22	BEF	3.5 (13)	5.05 (7)	4 (20)	6.5	13.2
ZBS3	E	4 (2)	5.12 (2)	4 (20)	2.3	39.8

## Data Availability

Detail of data will be provided on request.

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
