# Peer review of "Selection and Development of Nontoxic Nonproteolytic Clostridium botulinum Surrogate Strains for Food Challenge Testing"

_foods, 2022, doi:10.3390/foods11111577_

Round 1

Reviewer 1 Report

Selection and development of nontoxic nonproteolytic Clostridium botulinum surrogate strains for food challenge testing

This manuscript describes the authors’ examination of bacterial strains which mimic nontoxic nonproteolytic Clostridium botulinum in order to model the behavior of C bot in various foods.  Clostridium botulinum is a safety hazard as it produces a dangerous toxin.  It is important to study its behavior in foods to better understand foodborne botulism.  The creation of a good substitute is quite an advancement as it allows for greater accuracy in food studies in a safe manner.  Through their thorough characterization of a large number of strains, the authors identified several strains which could be used in varied conditions, representing a vast improvement for group II food studies.

The experiments appear to be designed appropriately, with proper controls.  I was quite impressed with the level of detail provided for the experiments. I was also impressed with the authors’ combination of the use of old techniques with new ones.  The authors seem very adept at using a wide variety of tools to answer questions and I enjoyed seeing a combination of various methods to come to a final conclusion.  The conclusions are not overreaching and are very appropriate based on the results.  The data support the authors’ conclusions.

I would therefore recommend one very minor comment.  Growth of C bot is a dangerous activity, and I would appreciate some mention of the hazardous nature of this activity as well as what safety measures were used by the authors to combat the dangerous nature of the work.  Section 2.2 of the Materials and Methods section should contain information on engineering controls (BSL-2, BSC, etc) which were used.

Author Response

Response to Reviewer 1

We thank the reviewer for the appreciation for our work and the thorough review. We have added some extra information regarding biosafety.

This manuscript describes the authors’ examination of bacterial strains which mimic nontoxic nonproteolytic Clostridium botulinum in order to model the behavior of C bot in various foods.  Clostridium botulinum is a safety hazard as it produces a dangerous toxin.  It is important to study its behavior in foods to better understand foodborne botulism.  The creation of a good substitute is quite an advancement as it allows for greater accuracy in food studies in a safe manner.  Through their thorough characterization of a large number of strains, the authors identified several strains which could be used in varied conditions, representing a vast improvement for group II food studies.

The experiments appear to be designed appropriately, with proper controls.  I was quite impressed with the level of detail provided for the experiments. I was also impressed with the authors’ combination of the use of old techniques with new ones.  The authors seem very adept at using a wide variety of tools to answer questions and I enjoyed seeing a combination of various methods to come to a final conclusion.  The conclusions are not overreaching and are very appropriate based on the results.  The data support the authors’ conclusions.

I would therefore recommend one very minor comment.  Growth of C bot is a dangerous activity, and I would appreciate some mention of the hazardous nature of this activity as well as what safety measures were used by the authors to combat the dangerous nature of the work.  Section 2.2 of the Materials and Methods section should contain information on engineering controls (BSL-2, BSC, etc) which were used.

  • We added two sentences at the end of the material and methods section about the biosafety measures that were taken in order to work with (material that could contain) potentially toxic C. botulinum strains.

‘All experiments described in this study were performed in a Biosafety level (BSL) 2 environment, but using BSL 3 practices when the possible presence of toxic C. botulinum strains (for environmental samples and enrichment cultures) or of botulinum toxin genes (for pure cultures) had not been excluded. Once the absence of the toxin genes had been demonstrated by whole genome sequencing, strains were handled using BSL2 procedures. This approach was formally approved by the Service for Biosafety and Biotechnology of the Scientific Institute for Public Health (Sciensano, document number SBB 219 2014/0018).’

Reviewer 2 Report

This is a very nice piece of work that has been done well. There are a few small remarks about the manuscript. However there are major serious concerns about the context of the work and how the strains might be used. These concerns must be addressed. It will be necessary for the revised manuscript to reviewed a second time.

Recommendations/guidelines in many countries mandate that food challenge studies with Clostridium botulinum require botulinum neurotoxin testing. Enumeration of bacteria is not enough. The advances described in this manuscript need to be placed clearly in this context. This could be done by stating that the strains developed can be used for initial screening or preliminary work in the same way that C. sprorgenes strains are sometimes used for group I C. botulinum, but they will never replace toxin-forming strains and measurement of botulinum toxin in final food challenge studies of a product.

Safety and security issues also need to be addressed. In which laboratories would these strains be used? Would they be used in a factory environment? Are there any safety or security concerns? Are there regulatory requirements? What is the possibility that the toxin gene could be intentionally introduced or accidently acquired (note that there is ample evidence of horizontal transfer of the neurotoxin gene in this species)? Would the authors provide the strains to any labs anywhere in the world?

Moreover, given the severity of the comments above it will be necessary for the revised manuscript to be subject to a second review. A small number of small remarks are made:

1) L38 – lower human oral lethal doses (1ng/kg) are more frequently reported

2) L43 – Serotype H is not considered a serotype. Only seven serotypes

3) Table 3 – A more common practice is not to mention the producer when products are tested (as it might be interpreted as criticism or endorsement of the products). Remove “brand” and details from Table 3.

4) Figure 2 is called “Phylogenetic trees of group II nontoxic and toxic strains” but three strains (ZBS2, CMCC3677 and ZBS15) are labeled as belonging to a “new cluster” and this might be interpreted as a new cluster of group II. Correctly, the text says otherwise, and it would be less confusing if it was called “non-group II cluster” or the like.

5) The physiological tests would have been better if there had been a direct comparison with toxic strains.

6) L340 – It is widely accepted that group II strains grow up to 5% NaCl. It is wrong to suggest there is any controversy about whether the strains grow above 3.5% NaCl. The sentence starting “However, there is” should be removed.

7) Figure 3 – it would help the reader to understand better how the D-values were calculated if example thermal death curves at 75C and 85C were added to the main manuscript.

8) Table 9 – given that challenge studies are often carried out at 4C, it seems essential that strains grow at 4C. Could the cocktail be increased to include more strains? If not, then note as a potential issue.

9) The supplementary material was not available to review.

Author Response

Response to reviewer 2

We thank the reviewer for the appreciation for our work and also for the thorough review and useful suggestions. We have carefully addressed the points one by one and made a number of adjustments to the manuscript accordingly.

This is a very nice piece of work that has been done well. There are a few small remarks about the manuscript. However there are major serious concerns about the context of the work and how the strains might be used. These concerns must be addressed. It will be necessary for the revised manuscript to reviewed a second time.

Recommendations/guidelines in many countries mandate that food challenge studies with Clostridium botulinum require botulinum neurotoxin testing. Enumeration of bacteria is not enough. The advances described in this manuscript need to be placed clearly in this context. This could be done by stating that the strains developed can be used for initial screening or preliminary work in the same way that C. sprorgenes strains are sometimes used for group I C. botulinum, but they will never replace toxin-forming strains and measurement of botulinum toxin in final food challenge studies of a product.

  • We added some nuance in the introduction and conclusion regarding this issue. More specifically, we clarified that the developed surrogate strain cocktail facilitates scientific experiments and large scale challenge testing, but that the botulinum neurotoxin assay may still need to be performed before products can be commercialized, depending on local legislation or guidelines.

Safety and security issues also need to be addressed. In which laboratories would these strains be used? Would they be used in a factory environment? Are there any safety or security concerns? Are there regulatory requirements? What is the possibility that the toxin gene could be intentionally introduced or accidently acquired (note that there is ample evidence of horizontal transfer of the neurotoxin gene in this species)? Would the authors provide the strains to any labs anywhere in the world?

  • The above questions relate to potential biosafety and security issues of the use of the strains in challenge testing by third-party users like food producers, commercial laboratories etc. While we agree that these issues are important, we believe they are not the primary subject of our work, which is in the first place a scientific contribution. We address the different points below in this rebuttal, and we have added some brief comments in the conclusions section of the manuscript to make the readers aware of safety issues that have to be addressed before the strains can be used.
  • With regard to neurotoxin-related biosafety and biosecurity issues, the strains developed in our work are in fact well comparable to C. sporogenes, the nontoxic equivalent of group I C. botulinum. C. sporogenes is classified as a BSL 1 organism (i.e. non-pathogenic), is not a dual use agent, and has a long history of use for challenge testing when the risk of group I C. botulinum in foods has to be assessed. To our humble opinion, there is little knowledge in how the BoNT-gene cluster is taken up by C. botulinum Group I, Group II, C. sporogenes, C. baratii or C. butyricum. For sure all these species occur as toxigenic strains harboring the BoNT-gene cluster and as non-toxigenic strains. The question about the risk taking up the BoNT-gene cluster in a laboratory environment where only non-toxigenic strains are cultured seems to neglectable. We are not aware of a single incident describing the conversion of a non-toxigenic strain (outside of C. botulinum Group III/C. novyi)..
  • In view of the above, we do not see any reason to apply specific restrictions to the distribution of the strains to other parties, and we will follow the European and Belgian legislation, and the policy and procedures of our University’s legal department herein. Evidently, the strains should be used only in a contained environment (one reason being that they carry an introduced antibiotic resistance marker) and not in a factory environment.
  • We want to point this reviewer also to the additional information regarding biosafety measures taken in the lab during the work described in this manuscript (question raised by reviewer 1).

Moreover, given the severity of the comments above it will be necessary for the revised manuscript to be subject to a second review. A small number of small remarks are made:

1) L38 – lower human oral lethal doses (1ng/kg) are more frequently reported

Agreed, and statement adjusted.

2) L43 – Serotype H is not considered a serotype. Only seven serotypes

We thank the reviewer for pointing this out. It is indeed correct that serotype H is also recognized by antisera against type A. On the other hand, a novel type X  toxin was reported recently (https://doi.org/10.1038/ncomms14130) which has a unique substrate profile and is not recognized by antisera against the other seven serotypes. We have updated the information accordingly.

3) Table 3 – A more common practice is not to mention the producer when products are tested (as it might be interpreted as criticism or endorsement of the products). Remove “brand” and details from Table 3.

Thanks for the suggestion, the specific details are removed.

4) Figure 2 is called “Phylogenetic trees of group II nontoxic and toxic strains” but three strains (ZBS2, CMCC3677 and ZBS15) are labeled as belonging to a “new cluster” and this might be interpreted as a new cluster of group II. Correctly, the text says otherwise, and it would be less confusing if it was called “non-group II cluster” or the like.

Thanks for the remark, the figure has been adapted.

5) The physiological tests would have been better if there had been a direct comparison with toxic strains.

We agree, of course, but this would have implied a considerable amount of additional work also requiring additional biosafety measures, and therefore we decided not to do this.

6) L340 – It is widely accepted that group II strains grow up to 5% NaCl. It is wrong to suggest there is any controversy about whether the strains grow above 3.5% NaCl. The sentence starting “However, there is” should be removed.

We changed the sentence to ‘However, some studies also report lower experimental maximum values, and these differences may relate to the different incubation times used [16], [46]. Possibly, extending the incubation time could result in higher salt tolerance values for our strains as well.’

7) Figure 3 – it would help the reader to understand better how the D-values were calculated if example thermal death curves at 75C and 85C were added to the main manuscript.

Since the inactivation curves highlighting the selected data points for all the strains are in the supplementary materials, the readers will have access to them, and we prefer not to duplicate such curves in the main manuscript.

8) Table 9 – given that challenge studies are often carried out at 4C, it seems essential that strains grow at 4C. Could the cocktail be increased to include more strains? If not, then note as a potential issue.

As can be seen in Table 9, three out of five strains from the cocktail are able to grow at 4°C (VAP51, ME22, ZBS3). In the composition of the cocktail, we have chosen to include not only strains that grow at 4°C, but also one or more strains that grow at high salt and at low pH. None of the strains is a champion in all these properties. Furthermore, challenge testing protocols commonly recommend the use of a variable temperature profile, including also a certain time at temperatures higher than 4°C (e.g. 7°C or even 12°C). Also, this manuscript reports about the development of a particular set of strains for use in challenge studies as a proof-of-concept, but this set can be adjusted depending on the design and purpose of the challenge study, for example to include more strains that grow at 4°C.

9) The supplementary material was not available to review.

We regret that this happened, since the supplementary material was uploaded together with the main manuscript file upon submission of the manuscript. We have notified the journal staff about the issue with the submission of the revised manuscript.
